# Environmental Substances Associated with Alzheimer’s Disease—A Scoping Review

**DOI:** 10.3390/ijerph182211839

**Published:** 2021-11-11

**Authors:** Hanna Maria Elonheimo, Helle Raun Andersen, Andromachi Katsonouri, Hanna Tolonen

**Affiliations:** 1Department of Public Health and Welfare, Finnish Institute for Health and Welfare (THL), Mannerheimintie 166, 00271 Helsinki, Finland; hanna.tolonen@thl.fi; 2Clinical Pharmacology, Pharmacy and Environmental Medicine, Department of Public Health, University of Southern Denmark, DK-5000 Odense C, Denmark; hrandersen@health.sdu.dk; 3Human Biomonitoring and Control of Industrial Products Laboratory, State General Laboratory, Ministry of Health, P.O. Box 28648, Nicosia 2081, Cyprus; akatsonouri@sgl.moh.gov.cy

**Keywords:** Alzheimer’s disease (AD), chemical exposure, pesticides, mercury (Hg), cadmium (Cd), arsenic (As), lead (Pb), HBM4EU

## Abstract

Alzheimer’s disease (AD) is the most common form of dementia, prevalent in approximately 50–70% of the dementia cases. AD affects memory, and it is a progressive disease interfering with cognitive abilities, behaviour and functioning of the person affected. In 2015, there were 47 million people affected by dementia worldwide, and the figure was estimated to increase to 75 million in 2030 and to 132 million by 2050. In the framework of European Human Biomonitoring Initiative (HBM4EU), 18 substances or substance groups were prioritized for investigation. For each of the priority substances, a scoping document was prepared. Based on these scoping documents and complementary review of the recent literature, a scoping review of HBM4EU-priority substances which might be associated with AD was conducted. A possible association between risk of AD and pesticides was detected. For mercury (Hg), association is possible but inconsistent. Regarding cadmium (Cd) and arsenic (As), the results are inconsistent but inclined towards possible associations between the substances and the risk of disease. The evidence regarding lead (Pb) was weaker than for the other substances; however, possible associations exist. Although there is evidence of adverse neurological effects of environmental substances, more research is needed. Environmental chemical exposure and the related hazards are essential concerns for public health, and they could be preventable.

## 1. Introduction

To understand the signs, symptoms and characteristics of AD, an overall term of dementia needs to be first elaborated. Dementia is an umbrella term for various diseases, including AD, which affect memory, thinking, behavior and emotion of a person leading to obstacles in daily living [1]. These are diseases which are regarded as chronic or progressive in nature including degeneration of cognitive function. Even though memory-affecting diseases are prevalent mainly in the elderly, they should not be considered as a normal part of ageing. The overall disease group of dementia is caused by different diseases and injuries influencing the brain, and the World Health Organization (WHO) considers dementia as a public health priority [2]. These diseases cause a significant health and economic burden worldwide. Approximately 47 million people worldwide had dementia in 2015, and the number is expected to nearly triple to 132 million by 2050. Interestingly, the biggest increments in incidents are detected nowadays in low- and middle- income countries. The worldwide cost of dementia was calculated to 818 billion USD in 2015, and these costs were estimated to increase to two trillion dollars by 2030. Of these costs, direct medical costs account for approximately 20%, whereas direct social sector costs and informal care costs cause both approximately 40% [3].

Many forms of dementia exist, and AD is the most common form of dementia attributing to 60–70% of cases. AD, like dementia, is a disease affecting mostly the elderly, and the diagnoses are doubled every 5 years after the age of 65 years [2,4,5]. A so-called ‘sporadic’ or ‘late-onset’ form is the most common type of AD. However, mutations in one of three genes, amyloid precursor protein (APP), presenilin 1 (PSEN1) and presenilin 2 (PSEN2), cause a rare (<0.5%), ‘familial’ or ‘early-onset’ form of AD (fAD). In the familial form of AD, symptoms are prevalent earlier than in the sporadic form, between the ages of 30 and 50 years [6]. Both genetic and environmental factors are supposed to contribute to the common start of sporadic or late-onset AD. Approximately 70% of AD risk is thought to be caused by genetic factors [4,7]. It is suggested by WHO [2], according to various studies, that the risk of dementia, and thus also AD, can be reduced by engaging in regular exercise, avoiding smoking and excessive use of alcohol, and by maintaining healthy weight, diet, blood pressure, cholesterol, and blood sugar levels. Besides age, the risk of developing the disease is increased with depression, low educational status, social isolation, and cognitive inactivity.

Every person with AD is affected differently, but because AD is a progressive disease, it contributes to the need of higher level of care of those who are having the disease for longest [8]. The ‘Consortium to Establish a Registry for Alzheimer’s disease’ (CERAD) has suggested six years as a mean time of disease progression from mild to severe case by using the ‘Clinical Dementia Rating Scale’ (CDR) [9]. Dementia, including AD, is a major contributor to disability and dependency of older people, and it is evident that the disease has both social and economic consequences. The burden of disease affects the people having it, their families, and caretakers as well as societies [2].

In addition to the most recognized risk factors of AD, some environmental substances are shown to be associated with adverse effects on cognitive functions and development of AD [10].

Human exposure to environmental chemical pollutants and the possible effects in the body can be measured directly in human fluids and tissues using ‘human biomonitoring’ (HBM). The ‘European Human Biomonitoring Initiative’ (HBM4EU) develops and advances HBM as a tool for health-related environmental monitoring and scientific policy-support for safer chemicals management by generating evidence of the actual exposure of citizens to chemicals and the possible health effects. In the frame of HBM4EU, the exposure of Europeans to 18 groups of chemical substances and their potential health effects are investigated [11]. This scoping review aims to present an overview of the latest epidemiological research evidence of the HBM4EU priority substances and their possible associations with AD. A disease-oriented approach is applied, and we are concentrating on investigating the risk which environmental substances are posing on human health, more specifically on the risk of AD.

In this review, the risk of AD associated with these specific environmental substances is discussed. The review aims to present an overview of their possible associations with AD in order to provide information for clinicians, stakeholders and policy makers. Our aim is to improve public health by increasing awareness of the health impacts of environmental substances and their possible role in contributing to health, social, and economic burden of AD. We believe that by applying a disease-oriented approach, the findings of this review can be useful in enhancing public health and in widening knowledge on the ever-increasing burden of environmental hazards. According to our knowledge, this is the first scoping review conducted on this subject.

## 2. Materials and Methods

This scoping review was performed in the concept of HBM4EU. HBM4EU is a joint activity of 30 European countries and the European Environment Agency (EEA), co-funded by the funding program of the European Commission, Horizon 2020. The HBM4EU project runs from 2017 until the end of 2021, and it aims to produce evidence of the citizens’ exposure to chemicals and the probable health impacts in order to reinforce policy making and preventive procedures [12].

A prioritization process in co-operation with policymakers, scientists, and stakeholders was conducted in order to select the substances to be studied under HBM4EU [13]. Eighteen substances and substance groups were selected based on two prioritization rounds: acrylamide, anilines, aprotic solvents, arsenic (As), benzophenones (UV filters), bisphenols, cadmium (Cd), chromium VI (Cr VI), flame retardants (FRs), lead (Pb), mercury (Hg), mycotoxins, per- and poly-fluorinated substances (PFASs), pesticides, phthalates and Hexamoll ^®^Dinch, polycyclic aromatic hydrocarbons (PAHs), chemical mixtures and emerging chemicals. Because pesticides comprise a huge group of diverse chemical substances, the following pesticides were prioritised in the HBM4EU project: pyrethroids (whole group), chlorpyrifos and dimethoate (organophosphate insecticides), fipronil (phenylpyrazole insecticides), and glyphosate (organophosphate herbicide). Substance-specific scoping documents have been prepared within the HBM4EU initiative including information about the substances and their relevance to policy [10].

The HBM4EU substance-specific scoping documents were used as a background material for this scoping review [10] because they include generally recognized or suspected adverse health effects associated with each prioritized substance group. From these scoping documents, we wanted to widen the knowledge further by concentrating on the adverse health effects regarding AD by offering a disease-oriented approach. Therefore, to investigate the environmental etiology of AD, a disease-orientated, complementary search of the literature was applied.

The literature search covered studies published during the last 10 years; however, the hand search included studies which were conducted earlier than this. The search was conducted in PubMed in April 2020 in order to collect information on the association between the HBM4EU priority chemicals and AD. The search was supplemented in August 2021 to cover the most recent literature of the subject. In the literature searches, different key words, such as ‘chemical exposure’, ‘environmental chemicals’ and each of the individual priority substances or group of substances and ‘AD’ or ‘dementia’, were applied. Regarding ‘pesticides’, the abovementioned specific pesticides were also applied in the search to supplement the search term ‘pesticides´. The search criteria were limited to epidemiological systematic reviews and meta-analysis introducing a possible association between AD and environmental substances. However, reviews, case-control, cross-sectional, or ecological studies were also included in order to elaborate on the findings, and hand-search was additionally used. All the selected studies were in English. The inclusion of the studies was conducted based on the titles and abstracts of the articles, and the author HME was responsible for selecting the studies to be included. Altogether 20 studies were selected, including 10 systematic reviews and meta-analyses, 1 synthesis of systematic reviews, 5 reviews, 1 cross-sectional study, 1 case-control study, 1 hypothesis, and 1 ecological study. PRISMA flow diagram was not applied because the review was conducted as a scoping review and not a systematic review. Experimental studies, i.e., in vitro or in vivo animal toxicity studies, were not separately considered, but if there were animal studies included in some of the reviews, this is mentioned in the result section. However, the results of the animal studies were mostly not reported in the results.

Based on the HBM4EU scoping documents and the supplementary literature search, the HBM4EU-prioritized chemicals associated with AD were identified to be pesticides, Cd, As, Pb, and Hg.

Scoping review methodology was applied in this review. Thus, no systematic review methods were used. The scoping review method can be useful in tackling broad questions and in investigating what has been presented on the subject in the earlier literature. The aim of the scoping review is not to produce a systematic summary of findings, or to assess the quality of the studies. On the other hand, it offers a noteworthy method for collating and assessing information preceding a systematic review [14,15,16].

## 3. Introduction of the Chemicals

A short introduction of the relevant HBM4EU-priority chemicals, which are associated with AD, is provided including pesticides, Cd, As, Pb, and Hg. The reader is referred to the HBM4EU scoping documents for a thorough presentation on current knowledge about each of the chemicals, including information about human exposure to them, possible health effects, knowledge gaps, open policy questions and activities under HBM4EU in support of answers to these questions [10].

Table 1 presents the matrices (e.g., human fluids and tissues) which have been demonstrated to be suitable in the frame of HBM for assessment of human exposure to the chemical substances of concern. However, not all the selected studies have used HBM in their analyses.

### 3.1. Pesticides

Pesticides are chemical compounds which are applied in order to remove pests including insects, rodents, fungi, and unwanted plants and weeds. Pesticides are used in both public health and agriculture; with the help of pesticides, vectors of diseases and pests that harm the crops are eliminated. Pesticides pose potentially toxic characteristics on humans by manifesting both acute and chronic health effects. More than 1000 pesticides are used worldwide with each of the pesticides having different properties and toxicological profile. The toxicity depends on the functions and other factors of the pesticides; e.g., insecticides are often more acutely toxic to humans than herbicides and fungicides [22,23].

Pesticides are classified according to their function (e.g., insecticides, herbicides, and fungicides) and chemical composition of the active ingredients. The classification of insecticides includes five main chemical groups: organochlorines (OCs), organophosphates (OPs), carbamates, neonicotinoids, and pyrethroids, but also some compounds belonging to other chemical classes are used as insecticides. Most insecticides target the nervous system of insects, and due to similarities in brain function, they also have neurotoxic properties in nontarget organisms including humans. Besides insecticides, some herbicides and fungicides also possess neurotoxic properties [24].

Pyrethroids form one of the major classes of insecticides in the European Union (EU) and worldwide. In relation to other major classes of insecticides, such as OPs, pyrethroids pose lower acute toxicity in mammals. However, epidemiological studies of pyrethroids are scarce, and therefore a complete picture of the potential health impact of low-level environmental and dietary exposure cannot be performed [10].

Humans are exposed to pesticides via ingestion, inhalation, or penetration through skin. The most vulnerable populations include pregnant women/fetuses, infants, and children. The general population is exposed to pesticides from residues in food and drinking water from residential use and from living close to pesticide treated agricultural areas. Occupational exposed groups, e.g., agriculture farm workers and pesticide applicators, are often more exposed than the general population [23,25]. Pesticides have been associated with a range of long-term health effects including neurodegeneration [25].

### 3.2. Cadmium

Cadmium is a naturally found heavy metal present at low levels in the environment. Both natural and anthropogenic sources exist. Natural sources include, e.g., erosion of parent rocks, volcanic eruptions, and forest fires, whereas anthropogenic sources consist of, e.g., plastics as color pigment and stabilizer, radiators, alkaline batteries, mining activities, and inappropriate waste disposal. The general population is exposed to Cd through food and drinking water, and inhaling tobacco poses a significant risk of exposure for smokers, making smokers a group of high exposure. Pregnant and postmenopausal women, elderly, and children are among vulnerable populations to Cd exposure [10,17,26,27].

Cadmium manifests many adverse health effects on humans; epidemiological studies among general populations have been able to show statistically significant associations with health effects even at low exposure levels. It should also be recognized that there are no safe Cd exposure levels [26].

### 3.3. Arsenic

Arsenic is a naturally occurring environmental toxicant present ubiquitously, and soil and drinking water contaminated by As is a global health threat. The main use of As and its compounds consists of alloys of lead, and As is a common n-type dopant in semiconductor electronic devices. Compared to organic As, the inorganic form, iAs, is extremely toxic, and in fact the risk assessments and recommendations are based on the iAs-levels. Humans are exposed to As via air, food, and water. Children are the most vulnerable and sensitive group of As exposure, and also occupational exposure in industries of gold mining and smelting operations poses a risk for workers. Among other health characteristics, As can be neurotoxic [10].

### 3.4. Lead

Lead is a toxic metal, which occurs naturally in the Earth’s crust. Environmental contamination, human exposure, and significant public health problems have occurred due to its widespread use worldwide. Lead and its compounds are used in various man-made substances such as petrol additives, Pb-based paints, Pb-pipes, and solders as well as Pb-soldered cans and batteries. Regarding the toxicity of Pb, it is noteworthy that it is a cumulative toxicant affecting several body systems including the nervous system. Lead is especially harmful during rapid brain development in young children and fetal life (i.e., pregnant women). Both occupational and environmental sources cause exposure to humans, and the main routes of exposure are inhalation of Pb particles and ingestion of Pb-contaminated dust, water and food, as well as trans-placental Pb release into blood. Among other health effects, Pb can cause cognitive effects in humans. No safe levels of Pb exposure exist [10,28].

### 3.5. Mercury

Mercury is a toxic metal found worldwide in air, water, and soil. It derives both from natural and anthropogenic sources. The three main forms of Hg are elemental (or metallic), inorganic and organic. Elemental Hg, also known as “quicksilver”, is acquired mainly from the refining of Hg sulfide in cinnabar ore and is also typically used in human activities (e.g., electrical equipment and dental amalgams). The anthropogenic use of Hg causes burden to the atmosphere and presents a considerable risk to human health and the environment. Inorganic Hg compounds occur when Hg combines with other elements such as chlorine, sulfur, and oxygen, and these compounds are used, e.g., in the production of batteries, polyvinylchloride, and pigments. The different forms of Hg are not equal in their harmful and toxic properties. The most poisonous form of Hg is the organic form, methylmercury, which is composed when inorganic Hg is methylated or merges with organic agents. Exposure to inorganic Hg happens via occupational exposure and to organic, methylmercury, via the diet (especially through the consumption of fish and shellfish) [10,29].

Because Hg is ubiquitous in the environment, virtually all humans are exposed to it at various degrees. The most common exposure routes are dermal, inhalation, oral, and transplacental. Mercury is a global threat to human health and the environment, and adverse health effects happen even at low levels of exposure. Especially fetuses, newborn babies, and children are regarded as vulnerable groups and sensitive to adverse effects of Hg exposure. Mercury compounds can cause irreversible damage to the central nervous system in humans because elemental and methylmercury are regarded as toxic to the central and peripheral nervous systems. Exposure to different Hg compounds can cause neurological, behavioral, reproductive, and developmental disorders. Furthermore, methylmercury compounds have been linked to cancer [10,29].

## 4. Results

The results of the selected studies are presented in the Table 2. The HBM4EU priority substances mentioned in the studies were the ones selected for the review. If other than the selected priority substances were investigated in the studies, they are not reported. The same applies for the diseases selected; in some of the studies, there were also other diseases than dementia or AD introduced, but in this review only the results concerning dementia, AD, and cognitive functions are presented.

### 4.1. Pesticides

Based on our scoping review, the evidence regarding the association between exposure to pesticides and risk of AD was rather strong. The strongest evidence was seen for occupational pesticide exposure.

Several systematic reviews and/or meta-analyses on AD and pesticides were found, and, not surprisingly, there was some overlap between the included studies. Because these studies contribute to a great extent to the overall conclusions, they are briefly described. A population-based, longitudinal study by Tyas et al., 2001 [50], demonstrating significantly increased risk of AD for occupational exposure to pesticides such as fumigants and defoliants (RR 4.35; 95% CI 1.05–17.90), was included in six systematic reviews and meta-analyses [30,32,33,34,36,37]. A community-based study by Hayden et al., 2010 [51] reported significantly higher risk related to occupational exposure to OP insecticides and nonsignificantly higher risk related to OCs. There were increased risks among pesticide-exposed individuals for all-cause dementia (hazard ratio (HR) 1.38; 95% CI 1.09–1.76), and for AD (HR 1.42; 1.06–1.91). The OP associated exposure caused a slightly higher risk of AD (HR 1.53; 1.05–2.23) than OCs (HR 1.49; 0.99–2.24). This study by Hayden et al., 2010 [51] was included in six systematic reviews and meta-analyses [30,31,32,33,34,36].

A case-control study by Richardson et al., 2014 [52] reported higher risk related to high serum concentration of DDE (metabolite of the OC DDT). The levels of DDE in serum were 3.8-fold higher in AD patients (mean 2.64 ng/mg cholesterol) in comparison to control participants (mean 0.69 ng/mg cholesterol; *p* < 0.001). Furthermore, the highest tertile of DDE levels was associated with an OR of 4.18 for increased risk for AD (95% CI 2.54–5.82, *p* < 0.001) and lower MMSE scores (−1.605; range −3.095 to −0.114; *p* < 0.0001). This study by Richardson et al., 2014 [52] was included in three studies [30,31,36]. Finally, a population-based case-control study, The Canadian Study of Health and Aging, 1994 [53], showing an OR of 1.58 (95% CI 0.81–3.10) for the risk of AD in relation to pesticides and fertilizers, was included in six studies [30,31,32,34,36,37].

The epidemiological evidence of an association between exposure to pesticides and risk of AD was widely supported, and a 30–50% increased risk of the disease was detected with exposure to pesticides [30,31,32]

Furthermore, in the systematic review by Killin et al., 2016 [33], two included reviews concluded that exposure to pesticides was associated with an increased risk of dementia. However, other included studies showed no associations, or had mixed findings. Olayinka et al., 2019 [34] had similar, rather mixed findings in their systematic review. However, in three out of six studies, statistically significant associations between pesticide exposure and AD were detected.

Krewski et al., 2017 [35] conducted a synthesis of systematic reviews by investigating risk factors behind 14 neurological conditions including AD. According to their conclusion, environmental exposure to pesticides may be associated with an increased risk of AD, and at least one systematic review rated of moderate/high quality has reviewed the available evidence and published a peer-reviewed report showing that a credible relationship exists.

Mostafalou and Abdollahi, 2017 [36] were able to demonstrate in the systematic review that most of the neurodegenerative disorders investigated, including AD, were associated with exposure to pesticides, and most notably to insecticides and herbicides, especially OPs and OCs. One ecological study included indicated that the prevalence of AD was two times higher among people living in the areas using more pesticides compared to others [54].

In the systematic review by Santibáῆez et al., 2007 [37], increased and statistically significant associations between AD and pesticides exposures were detected. To a wider category of pesticides/fertilisers, a smaller and nonsignificant association was detected (an adjusted relative risk (aRR) 1.45; 95% CI 0.57–3.68) [50]. In men, occupational exposure to pesticides showed a RR of 2.39; 1.02–5.63 [55]. Furthermore, one included case-control study [56] detected an unadjusted RR of 2.54 (0.41–27.06) for OPs.

Sharma et al., 2020 [42] reviewed available evidence of human neurotoxicity caused by toxic chemicals and concluded that some pesticides have been associated with AD.

Most of the studies on pesticides included in the systematic reviews evaluated the risk for AD associated with exposure to generic pesticides. However, a few of the studies within the included reviews presented information on more specific groups of pesticides and found significantly higher risk among individuals occupationally exposed to OPs [51] and pesticides used as fumigants/defoliants [50], and among individuals with elevated serum concentration of DDE which is a metabolite of the OC insecticide DDT [52].

### 4.2. Cadmium

The evidence regarding Cd exposure and AD is inconsistent and limited, but nevertheless, possible associations exist. There is a scarcity of epidemiological data, which hinders drawing of a thorough conclusion on the role of Cd in relation to the disease.

Xu et al., 2018 [38] demonstrated significantly elevated circulatory levels of Cd in AD patients compared to controls in their quantitative meta-analysis and systematic review. Cicero et al., 2017 [39], on the other hand, were unable to find conclusive data on Cd exposure in their systematic review. Sharma et al., 2020 [42] reviewed effects of environmental toxicants on neuronal functions and were able to highlight the impact of ubiquitous heavy metals on human health and how these metals increase neuronal dysfunctions including AD. However, the role of Cd was not highlighted in the review. Yang et al., 2018 [44] were unable to demonstrate the association between Cd and AD in their case-control study. Furthermore, in the ecological study, Li et al., 2020 [45] concluded that there was no association between soil Cd concentration and AD mortality. According to the review by Bakulski et al., 2020 [48], in human aging studies, Cd might be associated with decreased cognitive function and clinical AD specifically. However, the pathophysiologic link between environmental Cd exposure and AD was unclear.

### 4.3. Arsenic

The role of As in the risk of AD is unclear and scarcely studied, and formulating a conclusive observation is difficult.

The review by Gong and O’Bryant, 2010 [40] introduced a mere hypothesis of As exposure and AD instead of making a real synthesis of available evidence. Both animal and epidemiological studies were included. Arsenic exposure was associated with amyloid, vascular, and inflammatory hypotheses of AD. In the systematic review by Cicero et al., 2017 [39] contradictory evidence was introduced; in one epidemiological study, higher soil As levels were associated with higher rates of AD in European countries, but no evidence has been detected when measuring As levels from biological samples. Killin et al., 2016 [33] found rather weak evidence regarding the association between As exposure and AD/dementia according to two studies in their systematic review. Sharma et al., 2020 [42] strengthened the evidence of the effects of heavy metals, including As, in neuronal disorders such as AD. In the case-control study by Yang et al., 2018 [44] the AD risk of study participants with high urinary InAs% or low DMA% was significantly increased. People with relatively high InAs% or low DMA% were considered to have poor As methylation capability and increased body retention of As. In the cross-sectional study by Wang et al., 2021 [43], the cognitive-impaired group had significantly higher hair As concentrations and the prevalence of As poisoning (arsenicosis) than the cognitive-normal group. HairAs concentrations were negatively correlated with MMSE scores, and arsenicosis seemed to be a risk factor for cognitive impairment. According to the ecological study by Li et al., 2020 [45], AD mortality was increased by soil As concentration.

### 4.4. Lead

Regarding Pb exposure and AD risk, the evidence is conflicting and from weak to moderate.

In the systematic review and meta-analysis by Xu et al., 2018 [38], the blood Pb levels were reduced in AD patients compared to controls, and in the systematic review by Santibáῆez et al., 2007 [37], there was no evidence of the association between professional exposure to Pb and AD. Moreover, Killin et al., 2016 [33] found rather weak evidence regarding occupational Pb exposure and AD/dementia in their systematic review. Olayinka et al., 2019 [34] were unable to assess the effect of Pb exposure to AD due to low number of studies in their systematic review. Cicero et al., 2017 [39] came up with inconclusive findings in their systematic review; a possible association between blood Pb levels and AD was evaluated in several case control studies but without a significant increase in concentration (except for one study). Moreover, two studies on CSF showed conflicting results; in one, no difference, and in another, reduced concentration of Pb was detected. Furthermore, one study showed reduced Pb concentration in hair and no difference in nail samples. In the case-control study by Yang et al., 2018 [44], Pb exposure was not associated with AD. Similar results were detected in the ecological study by Li et al., 2020 [45] where there was no association between soil Pb concentration and AD mortality. However, in the review, Sharma et al., 2020 [42] were able to highlight the evidence of the association between heavy metals, such as Pb, and neuronal disorders including AD. Furthermore, Bakulski et al., 2020 [48] concluded in their review that Pb exposure was associated with lower cognitive status and longitudinal declines in cognition in older adults. Moreover, in the systematic review by Loef et al., 2011 [49], Pb had a potential role in the development of AD and as a risk factor for AD. Long-term Pb exposure was suggested to be associated with the cognitive decline in elderly.

### 4.5. Mercury

The findings of the contribution mechanism of Hg to the risk of AD are inconsistent to some extent. However, there is a possible association between cognitive functions/risk of AD and exposure to Hg. Furthermore, occupational exposure is of some concern.

The Hg levels measured from different biomonitoring matrices show varying results; measurements from blood, especially serum, have shown the most consistent findings of elevated Hg levels in AD patients compared to controls [38,39]. Nevertheless, even blood samples have given contradictory results, as well as hair samples [39,41]. CSF Hg concentrations showed no difference, and nail Hg concentrations were significantly lower in AD patients according to two studies [39].

An inverse relation between Hg excretion and cognitive functions in exposed workers was detected in the systematic review by Cicero et al., 2017 [39]. The levels of Hg in blood from AD cases compared to controls varied within the studies. There were no differences found based on Hg concentration in CSF, and in some studies, nail concentration of Hg was markedly reduced and concentrations in hair presented inconsistent results.

Olayinka et al., 2019 [34] concluded that the influence of Hg on AD risk was difficult to determine due to small number of studies in the systematic review. However, in the systematic review and meta-analysis by Xu et al., 2018 [38], it was shown that elevated Hg levels in the circulation may contribute to the progression of AD.

Mutter et al., 2010 [41] detected significant memory deficits in people exposed to inorganic Hg according to most of the studies in their systematic review. In some autopsy studies, there was evidence of elevated Hg levels in brain tissues of AD patients. Variations and inconsistencies of Hg levels between the different biomarkers were detected. According to the results, inorganic Hg might be associated with the development of AD, increase the pathological influence of other metals, and promote neurodegenerative disorders.

Sharma et al., 2020 [42] included Hg in their review of neurotoxic metals and concluded that these metals are associated with AD. However, in the case-control study by Yang et al., 2018 [44], Hg was not associated with AD risk. Furthermore, in the ecological study by Li et al., 2020 [45], there was no association between Hg soil concentrations and AD mortality.

Siblerud et al., 2019 [46] created a hypothesis identifying factors associated with AD and showed that all these factors could be explained by Hg toxicity. In the review by Azar et al., 2021 [47], it was demonstrated that Hg can contribute to the development of AD.

## 5. Discussion

The aim of this scoping review was to investigate and clarify which of the specific environmental substances are possibly affecting the risk of AD. The substances investigated were derived from the prioritization list of the HBM4EU and included pesticides, Cd, As, Pb, and Hg. The evidence was gathered based on the literature search limited to epidemiological systematic reviews and meta-analyses. Some additional studies were identified via hand-search, and in some occasions, reviews or studies with other designs were introduced. In some of the selected reviews, there were animal studies included, but the results from these studies were not emphasized in our results. In the search terms, both “dementia” and “Alzheimer’s disease” were used in order to cover the full spectrum of the disease.

Exposure to pesticides seems to be associated with the risk of AD rather strongly, and according to our review, exposure to pesticides was the strongest contributor to the risk of AD from the substances selected. The evidence regarding Hg is slightly inconsistent, but associations are possible. With As, the results are showing possible associations, but some inconsistencies between the studies can be seen. Furthermore, due to the lack of identified reviews, a few of the As studies identified were either case-control, cross-sectional, or ecological studies. With Cd the results are contradictory, but possible association between the substance and the risk of disease is detected. Furthermore, the evidence regarding Pb is conflicting and somewhat weaker compared to the other substances. However, even with Pb, a possible association with cognitive decline and AD exists.

In this review, pesticides included the strongest evidence regarding the risk of AD. The systematic reviews and meta-analyses of pesticides encompassed studies from Europe, UK, Middle East, Asia, Australia, South Africa, South and Central America, Canada, and US. Pesticides have been widely used in food production all over the world for decades. Some older and cheaper pesticides can remain in soil and water for years, and the most hazardous of these pesticides have been banned from agricultural use in the developed countries, while they are still widely used in many developing countries. The WHO suggests banning pesticides that are most toxic to humans and the pesticides that remain in the environment for longest. Moreover, maximum limits for pesticide residues in food and water are introduced in order to safeguard public health [23]. The EU pesticide laws are considered the strictest in the world, and a thorough science-based assessment of safety is needed before an active substance is approved by the European Commission. During the last decades, a rigorous review regarding the use of all the pesticide substances in the EU has taken place, and the number of approved active substances has been diminished by more than 50%. To give perspective, previously more than 1000 substances were on the market in the EU, but now only around 400 are approved [57]. However, many pesticides with neurotoxic properties are still in use both in the EU and worldwide.

The general population is exposed to a variety of different pesticides, and often both diet and occupational settings include exposure to a mixture of pesticides. Although the exposure level of the individual pesticides is low, exposure to several pesticides with similar mode of action or toxicity to the same target organs is likely to be cumulative and therefore more hazardous. Investigating the risk of pesticide exposure in detail can be challenging because epidemiological studies may provide evidence of adverse health effects regarding mixtures of pesticides without identifying individual pesticides or pesticide groups [10]. In this review, a few of the selected studies on pesticides included some individual pesticide classes, mainly OP and OC insecticides, but studies addressing other specific types of pesticides, e.g., pyrethroids or neurotoxic herbicides and fungicides, were lacking.

One finding from this review is that harmful effects of chemicals are often investigated and related to occupational settings, especially considering the exposure to pesticides. Thus, close attention should be paid to the health effects of certain occupational groups, such as pesticide applicators and farmers, because pesticide exposure seems to be linked to increased disease rates of AD [36,37]. Moreover, regarding the association of heavy metals exposure and the risk of AD, occupational exposure is of some concern, and the exposure levels of heavy metals are often researched in occupational settings [39,41]. To tackle the considerable health burden of work-related exposure requires launching of adequate safety procedures and protection mechanisms at work. The crucial role of occupational exposure demonstrates the need to carefully define the harmful threshold values of environmental chemicals. Additionally, substituting harmful substances with less harmful ones should be an essential part of the future safety policies at workplaces.

The EU chemicals legislation has been regulating the chemical sector and the industries using chemicals since the 1960s by covering the lifecycle of products and protecting the environment and human health from chemical hazards and risks. The first piece of chemicals legislation was the Dangerous Substances Directive launched in 1967 in order to safeguard public health and especially the health of workers exposed to dangerous substances. More recently, this directive has been improved and replaced by the Regulation on Classification, Labelling and Packaging of substances and mixtures (the CLP Regulation), and other pieces of legislation have been applied in monitoring hazardous chemicals in water, waste, fertilizers, pesticides, industrial activities, consumer products, and occupational settings [58]. The HBM4EU initiative has an important role in providing evidence of the actual chemical exposure of citizens to chemicals and the possible health effects with the aim to support policy making [11].

The evidence detected regarding the associations between the heavy metals Cd, As, Pb, and Hg and the risk of AD was somewhat inconsistent and weak. However, there were possible associations established, and obviously more research is needed to confirm these findings. Therefore, it should not be concluded that contribution of these substances to neurological diseases is completely null but rather tentative, and that the evidence is incomplete. With the help of further research, some of the possible associations may be strengthened, and some may be proven futile. According to Yan et al., 2016 [32], high-class cohort studies with prospective design are the most reliable in showing the associations between the exposure and the disease.

The findings of this review indicate that environmental substances may be associated with the increased risk of AD. However, the evidence gathered in here is by no means comprehensive. The scoping review methodology excludes formal quality assessment of studies [59], and this prevents us from making final judgements or thorough conclusions of the evidence, let alone conducting a formal synthesis of the studies.

The amount of epidemiological evidence of the association between chemical exposures and neurological disorders including AD is still rather scarce, limited, and inconsistent in order to draw definite conclusions, or to determine causal pathways. Often there are confounding factors included, and measurements matrices of chosen substances vary across the studies or even within the studies affecting reliability and validity of the studies, which obviously makes comparing and synthesizing the results challenging. Furthermore, the study populations are often small and the follow-up periods (if any) are short. The adjustments varied between the studies, and the results were mostly adjusted for age, sex, education, family history of dementia or AD, residency in either institution or community, alcohol consumption, and smoking. Because we were not conducting a meta-analysis, and availability of adjustment variables were not complete for all studies, we did not find it feasible in the scope of our study to do interpretation based on them.

One obvious limitation and weakness of this review deals with the fact that most of the selected studies were either reviews, systematic reviews or meta-analysis. Therefore, some of the original studies included in various reviews may have caused the results of these specific studies to be disproportionately emphasized because the same studies have been cited in different reviews. This obviously causes some bias on the results and hinders our making strong conclusions on this researched topic. Nevertheless, this scoping review demonstrated the potential association between certain environmental substances and AD.

Further research is highly encouraged for strengthening the findings and safeguarding of public health. High-quality epidemiological studies with large enough study populations, preferably with cohort design, are strongly called upon. Only by applying long-term monitoring, more comprehensive statements and possible causality mechanisms of the adverse health effects of the environmental substances can it be outlined. Aligned with the scoping review methodology [14,15,16,59], the need for more extensive research and development is justified according to this review.

Understanding of the underlying mechanisms of the associations requires a versatile approach. Bioaccumulation processes of substances as well as threshold values for adverse effects should be closely defined, if possible. As people are exposed to many environmental substances simultaneously, urgent research is needed to investigate the cumulative effects of mixed chemical exposure on human body and on burden of diseases. The economies of the developing countries are evolving, and people are exposed to ever increasing amounts of environmental substances causing a substantial risk not only on neurological and cognitive diseases but on other diseases as well. To conclude, high-quality multidiscipline research is strongly warranted for the sake of protecting citizens from the health hazards caused by environmental substances.

## 6. Conclusions

According to this scoping review, the strongest evidence was detected for pesticides regarding the risk of AD.Exposure to mercury is possibly associated with AD development, but results are inconsistent. Though results on cadmium and arsenic and AD are conflicting, they are possibly associated with the risk of disease. The evidence regarding lead is less strong, but also lead can contribute to cognitive decline and the risk of AD.The studies selected were often connected to occupational settings, and thus far, many of the chemical exposure studies have been conducted in relation to high exposure occupations.Exposure to pesticides, mercury, cadmium, arsenic, and lead may adversely affect cognitive abilities and contribute to the risk of AD. Further, high-quality epidemiological research is needed to confirm the findings.Because adverse environmental chemical exposures affect both workers in certain occupations and general public, protecting humans from chemical hazards is an essential public health issue.

## Figures and Tables

**Table 1 ijerph-18-11839-t001:** Biological fluids and tissues (‘matrices’) suitable for assessing human exposure to each investigated chemical substance.

Substance	Measurement Matrix
As [10]	Urine, blood, sometimes measured in hair, nails
Cd [10,17,18,19]	Urine: long-term accumulation and exposure, blood: short-term exposure, sometimes measured in hair, nails, saliva, breast milk, placenta, or meconium
Pb [10,18,20,21]	Blood: recent exposure, bone: long-term exposure, urine: long-term occupational exposure, sometimes measured in hair, nails, saliva, breast milk, placenta, or meconium
Hg [10,18]	Urine, blood, hair, nails, sometimes measured in breast milk or meconium
Pesticides [10,18]	Urine: short-term exposure, blood: body burden of persistent organochlorine pesticides, sometimes measured in hair: long-term exposure, or in breast milk, meconium, or placenta

**Table 2 ijerph-18-11839-t002:** Overview of the reviewed studies and their main results related to investigated chemicals and AD.

Study:	Included Countries:	Study Design:	Sample:	Chemical:	Exposure Assessment:	Outcome(s):	Results and Main Conclusions:	Summary *:
Aloizou et al., 2020 [30]	US, UK, Ecuador, South Africa, Costa Rica, Brazil, Egypt, Spain, Iran, France, India, Sweden, China, Chile, Gulf, Netherlands, Greece, Canada, Australia, Taiwan	Review	*N* of studies: 30 human studies (and 15 animal studies) Setting: cross-sectional, longitudinal, case-control, cohort, case cohort, and ecological studies Target group: occupational	pesticides	questionnaire: occupational and non-occupational, environmental and self-reported-pesticides exposures, and place of residence serum: cholinesterase, acetylcholinesterase, and OC pesticides levels plasma: butyrylcholinesterase (BuChE) and red blood cell acetylcholinesterase (AChE) activity and concentrations of 3 OC pesticides urine: TCP levels, DEP, DETP, DAP metabolites, and TCPy	cognitive functions, dementia, and AD	An overall indication from epidemiologic evidence supported an association between exposure to neurotoxic pesticides and cognitive dysfunction, dementia, and AD.	⇧
Gunnarsson and Bodin, 2019 [31]	Not specified	Systematic literature review and meta-analyses	*N* of studies: 31 studies on pesticides (4 on AD) and 14 studies on metals (3 on AD) Setting: cohort or case-control studies Target group: occupational	pesticides	Job Exposure Matrix (JEM), self-reported exposure (questionnaire/interviews), DDE concentration in serum, AD diagnosis, severity of AD measured by the Mini-Mental State Examination (MMSE) score, and interaction with apolipoprotein E (APOE4) status	AD	The relative risk (RR) for AD and pesticides was 1.50 (95% confidence interval (CI) 0.98–2.29). Exposure to pesticides increased the risk of AD by at least 50%.	⇧
Yan et al., 2016 [32]	Canada, France, US, Australia	Systematic review and meta-analysis	*N* of studies: 7 Setting: 3 cohort and 4 case-control studies Target group: occupational, AD patients, and healthy controls	pesticides	prospectively, self-report ever/never occupational exposure, retrospectively, proxy reports job history, risk factor information, and code into JEM	AD	A positive association between pesticide exposure and AD was detected: odds ratio (OR) = 1.34 (95% CI 1.08–1.67). The summary ORs from the crude and adjusted effect size studies were 1.14 (0.94–1.38, *N* = 7) and 1.37 (1.09–1.71, *N* = 5), respectively. All subgroup analyses showed that pesticide exposure was associated with an increased risk of AD but only half of them with statistical significance. A significantly increased risk detected in cohort studies: OR = 1.37 (1.08–1.75) but not in case-control studies: OR = 1.24 (0.78–1.97). The OR of the self-reported group was 1.37 (1.08–1.75), and the OR of the proxy-reported group was 1.24 (0.78–1.97) when stratified according to method used to access exposure.	⇧ ⇔
Killin et al., 2016 [33]	Canada, US, France, Netherlands, UK	Systematic review	*N* of studies: 60 (10 on pesticides, 2 on As, and 2 on Pb) Setting: 6 cohort, 2 cross-sectional, 2 review studies on pesticides, 2 cross-sectional studies on As, 1 cross-sectional, and 1 review study on Pb Target group: occupational, environmental, and regional	pesticides, Pb, and As	self-reported risk factor and exposure to pesticides by questionnaire, exposure to pesticides from residential history and census data, cumulative exposure to pesticides based on job history, comparison of annual aveage-adjusted mortality rates between two different locations (As), principal components analysis, and AD cases identified from a registry (Pb)	dementia, and AD	Quality of overall evidence regarding defoliants/fumigants was weak but regarding pesticides/fertilizers/herbicides/insecticides strong. Among the different chemicals investigated, the strongest evidence was found for pesticide exposure but findings were heterogenous. Evidence of associations between occupational exposure, Pb, As and dementia was rather weak.	⇔ for pesticides ⇩ for Pb and As
Olayinka et al., 2019 [34]	Sweden, UK, Canada, France, US, Australia, Turkey, Denmark, Taiwan, Finland, Japan	Systematic review	*N* of studies: 29 (6 on pesticides, 2 on Pb, and 1 on Hg) Setting: 11 cohort and 18 case-control studies Target group: occupational and environmental	pesticides and metals (Pb, Hg)	occupational and environmental exposure	AD	Significant evidence was detected of the association between pesticide exposure and AD, depending on the group of the pesticides; occupational exposure to OP pesticides, fumigants and defoliants was more significant than occupational exposure to pesticides such as herbicides and insecticides in relation to AD risk. Due to the low number of studies, the effect of Pb and Hg on AD risk was difficult to estimate.	⇧ ⇩ for pesticides ⇔ for Pb and Hg
Krewski et al., 2017 [35]	Not specified	Synthesis of systematic reviews	Studies on the risk factors associated with the onset and progression of 14 neurological conditions, including AD	pesticides	-	AD	Exposure to pesticides was associated with AD with sufficient evidence.	⇧
Mostafalou and Abdollahi, 2017 [36]	Not specified	Systematic review	*N* of studies: 448 (6 on AD) Setting: 2 case-control, 3 cohort, and 1 ecological studies Target group: occupational and environmental	pesticides	questionnaire, geographic information system (GIS), and level of pesticides in serum	AD	In cohort studies there were 1.4- and 2.4-times higher risk of AD in people occupationally exposed to any pesticides and 1.5 increased risk of AD with exposure to OP and OC compounds based on the longitudinal and prospective analysis of exposures.	⇧
Santibáῆez et al., 2007 [37]	Canada, France, US, England and Wales, Australia	Systematic review	*N* of studies: 24 (6 on pesticides and 6 on Pb) Setting: 21 case-control and 3 cohort studies Target group: occupational	pesticides and Pb	occupational exposure	AD	Increased and statistically significant associations between AD and pesticide exposure were observed in studies of greater quality and prospective design. No evidence was detected of association for Pb.	⇧ for pesticides ⇩ for Pb
Xu et al., 2018 [38]	Korea, Australia, Spain, Italy, China, Sweden	Quantitative meta-analysis and systematic review	*N* of studies: 42 (7 on Hg, 8 on Cd, and 10 on Pb) Setting: case-control studies Target group: AD patients and healthy controls	Cd, Hg, and Pb	levels of toxic metals assessed from the circulation (blood, serum/plasma)	AD	According to the meta-analysis significantly elevated circulatory levels of Hg: pooled standardized mean difference (SMD) = 0.55 (95% CI 0.15–0.95, *p* = 0.0073) and Cd (SMD = 0.62, 0.12–1.11, *p* = 0.0144) were detected in AD patients compared to controls. Regarding Pb, reduced circulatory levels detected in AD patients in comparison to controls (SMD = −0.23, −0.38 to −0.07, *p* = 0.0043). Elevated Cd- and Hg-levels in the circulation, especially in serum, may contribute to the progression of AD.	⇧ for Hg and Cd ⇩for Pb
Cicero et al., 2017 [39]	Not specified	Systematic review	*N* of studies: - Setting: case-control studies Target group: occupational, AD patients, and healthy controls	Pb, Hg, Cd, and As	blood/serum/plasma, cerebrospinal fluid (CSF), nail, and hair	AD, dementia, and cognitive functions	An inverse relation between Hg excretion and cognitive functions in exposed workers was detected. In four case-control studies higher blood Hg levels in AD patients compared to controls. However, in the other studies, lower concentrations of Hg (one study) or no difference (four studies). No differences in the CSF Hg concentration (two studies). Nail-Hg concentrations significantly lower in AD patients (two studies), and inconsistent findings in the hair-Hg examination (two studies). Data was inconclusive on Cd, As, and Pb.	⇧ ⇩ ⇔
Gong and O’Bryant, 2010 [40]	Not specified	Review	Both animal and epidemiological studies	As	Not specified	AD	As-exposure hypothesis was supported by a simple mechanism for AD development and progression in certain AD patients. As exposure is associated with amyloid, vascular, and inflammatory hypotheses of AD.	⇧
Mutter et al., 2010 [41]	Not specified	Systematic review	*N* of studies: 88 (incl. 8 animal studies) Setting: mainly case-control and comparative cohort studies, also cross-sectional studies Target group: occupational	Hg	brain tissue, blood, urine, hair, nails, and CSF	AD	Significant memory deficits in individuals exposed to inorganic Hg was found in 32 out of 40 studies. In some autopsy studies elevated Hg-levels detected in brain tissues of AD patients. Measurements of Hg-levels in blood, urine, hair, nails, and CSF were discrepant. Inorganic Hg may contribute to the development of AD, increase the pathological influence of other metals, and promote neurodegenerative disorders via disruption of redox regulation. The influence of inorganic Hg on the nervous system was weaker in epidemiological studies compared to animal and in vitro-studies.	⇧ ⇔
Sharma et al., 2020 [42]	Not specified	Review	The available data for human neurotoxicity due to toxic chemicals was collected	Pb, Hg, Cd, As, and pesticides	-	AD	Neurotoxic metals such as Pb, Hg, Cd, and As as well as some pesticides have been associated with AD because of their ability to produce senile/amyloid plaques and neurofibrillary tangles (NFTs)-the features behind the neuronal dysfunctions such as AD.	⇧
Wang et al., 2021 [43]	China	Cross-sectional study	*N*: 1556 adults (802 males and 754 females, an average age of 57.0 +/− 11.5 years). 321 (20.6%) participants in the cognitive-impaired (CI)-group and 1235 (79.4%) participants in the cognitive-normal (CN)-group.	As	cognitive function measured by Chinese version of MMSE, internal-As exposure measured by hair As-concentrations (HAs), and external-As exposure measured by the distance between the participant‘s location of residence and the Realgar Plant	cognitive impairment	The CI-group had a significantly lower MMSE score compared to the CN-group (16.6 +/− 5.47 vs. 26.3 +/− 2.81, *p* < 0.05). In the CI-group much higher HAs than in the CN-group with a statistically significant difference (0.27, 0.14–0.56 mg/kg vs. 0.20, 0.10–0.41 mg/kg, *p* < 0.05). The prevalence of arsenicosis (arsenic poisoning) was significantly higher in the CI-group compared to the CN-group (64.5% vs. 45.3%, *p* ˂ 0.05). A negative correlation between hair-As concentrations and MMSE scores was detected (correlation coefficient (r) = −0.151, *p* < 0.001). Arsenicosis was a risk factor for cognitive impairment (OR = 1.84, *p* ˂ 0.05).	⇧
Yang et al., 2018 [44]	Taiwan	Case-control study	*N*: 434 adults (170 AD patients and 264 controls, age ≥ 50 years, an average age of 73.65, standard deviation (SD) 8.46). A propensity-score-matched population of 82 AD patients and 82 controls.	Cd, Pb, Hg, and As	AD patients: clinical neuropsychological examination and cognitive-function assessments incl. MMSE and clinical dementia-rating scale. Blood levels of Cd, Pb, and Hg and urine levels of As levels.	AD risk	The AD risk of study participants with high urinary inorganic As (InAs%) or low dimethylarsinic acid (DMA%) was significantly increased (*p* ˂ 0.05), similar findings in the propensity-score-matched population. People with high median level of InAs% or/and a low median level of DMA% had approximately two- to threefold significant AD risk. Cd, Pb, and Hg were not associated with AD risk.	⇧ for As ⇩ for Cd, Pb, and Hg
Li et al., 2020 [45]	China	Ecological study	22 provinces and 3 municipal districts in mainland China	As, Pb, Cd, and Hg	As- concentrations in soil in 1990 obtained from the China State Environmental Protection Bureau and data on annual mortality of AD from 1991 to 2000 obtained from the National Death Cause Surveillance Database of China	As concentrations in soil and AD mortality	AD mortality was increased by soil As concentration, the Spearman correlation coefficient between As concentration and AD mortality was 0.552 (*p* = 0.004), 0.616 (*p* = 0.001), and 0.622 (*p* = 0.001) in the A soil As (eluvial horizon), the C soil As (parent material horizon), and the total soil As (A soil As + C soil As), respectively. Evidence was found of a possible causal association between As exposure and the death risk from AD. No association was detected between Pb, Cd, and Hg soil concentrations and AD mortality.	⇧ for As ⇩ for Pb, Cd, and Hg
Siblerud et al., 2019 [46]	Not specified	Hypothesis	-	Hg	70 factors were identified as occurring in AD; factors were investigated in relation to Hg exposure	AD	All 70 factors associated with AD were examined and all of them could be explained by Hg toxicity. The hallmark changes of AD: plaques, beta amyloid protein, neurofibrillary tangles, phosphorylated tau protein, and loss of memory could all be changes caused by Hg. Neurotransmitters such as acetylcholine, serotonin, dopamine, glutamate, and norepinephrine are inhibited in AD patients, and same inhibition occurs in Hg toxicity. It was strongly suggested that Hg can cause AD.	⇧
Azar et al., 2021 [47]	Not specified	Review	Both animal and epidemiological studies	Hg	-	AD	Hg was involved in the process of amyloid beta deposition and tau tangles formation, which contributed to the development of AD.	⇧
Bakulski et al., 2020 [48]	Not specified	Review	Both experimental and epidemiological studies, partly in occupational settings	Pb and Cd	Pb in blood or in bone (tibia or patella) and Cd in brain tissues (postmortem studies), in circulation concentrations (whole blood, serum, or plasma), or in urine	AD, dementia, and cognitive decline	Pb exposure was associated with lower cognitive status and longitudinal declines in cognition in older adults. Cd may be associated with decreased cognitive function and clinical AD specifically in human aging studies. The pathophysiologic link between environmental Cd exposure and AD was limited due to the uncertainty in Cd transport to the brain.	⇧ for Pb ⇔ for Cd
Loef et al., 2011 [49]	Not specified	Systematic review	*N* of studies: - Setting: both animal and epidemiological studies, human studies: longitudinal cohort, cross-sectional, and case-control studies, as well as case-report and post-mortem analysis Target group: e.g., occupational and non-occupational cohorts, AD patients, and healthy subjects	Pb	Pb in blood, bone (tibia, patella, or calcaneus), CSF, dentate gyrus, temporal cortex, or in protein fraction from cortical gray and subcortical white matter	cognitive decline, AD, and dementia	Pb had a potential role in the development of AD, and was a risk factor for AD. An effect of long-term Pb exposure on cognitive decline in elderly subject was suggested. A scarcity of conclusive studies including patients with validated AD diagnoses was detected.	⇧

* ⇧ association observed, ⇩ negative association or no association observed, ⇔ inconsistent associations/results observed.

## Data Availability

Not applicable.

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
