# Peer review of "Environmental Substances Associated with Alzheimer’s Disease—A Scoping Review"

_ijerph, 2021, doi:10.3390/ijerph182211839_

Round 1
Reviewer 1 Report
This review covers the possible association between Alzheimer’s disease (AD) and environmental substances. Environmental factors are emerging as key players in AD etiology and the contribution of these factors to the pathology of AD is widely acknowledged. It's a timely topic and there is a lot of interest in environmental factors related to AD, although it has also been covered in several reviews quite recently. Nevertheless, updated reviews always are welcome and I think this work will be a good addition to the body of literature on AD. I find the review easy to follow and sections are well organized and written and present the state of art well.
However, I strongly suggest changing the focus of the introduction section and abstract towards AD only, instead to dementia and AD, as the title of the review is “Environmental substances associated with AD – A scoping review”. Furthermore, I consider that the terms dementia and AD cannot overlap since AD is a form of dementia, existing many other forms of dementia with different pathological mechanisms. In addition, the most of the outcomes of the studies included in the table 2 are focused on AD, so changing the introduction would not affect the rest of manuscript.
On other hand, I miss a brief description of the molecular mechanisms involved in the AD pathology related to environmental substances. I mean to comment briefly the common mechanisms by which environmental contaminants could induce AD pathology (enhancing amyloid-β peptide or tau phosphorylation, oxidative stress, mitochondrial dysfunction…) leading to neuronal loss. For example, Pb has been reported to increase expression of APP mRNA as well as the number of amyloid plaques through increased expression of BACE1 and APP. It is possible that this is not the aim of this work, but I think that it would improve the quality of the review.
Minor points:
- Line 90: missing word “causing … on human health”
- Abbreviations described previously in Material and Methods section. Do not repeat again.
- Lines 206, 338: Cadmium (Cd)
- Lines 220, 355: Arsenic (As)
- Lines 230, 378: Lead (Pb)
- Lines 243, 402: Mercury (Hg)
- Line 438: Alzheimer’s disease (AD)
- Line 440
- Extra spaces between words:
- Lines 258, 336, 398, 429
- Line 379: ´´Regarding Pb exposure and AD risk, the evidence…´´ rather than ´´Regarding Pb the evidence…”
- Line 414: ´´The levels of Hg in blood from AD cases´´ rather than ´´The levels of Hg in blood in AD cases ´´
- Take care of the commas in the text. There are some commas that are missing and others that are left over. Some examples lines 326, 368, 372, 380, 397, 435, 513

Reviewer 2 Report
This manuscript is a generally well-written review of some of the literature on the relationship between environmental exposures and the development of dementia. There are some issues that should be addressed with the manuscript before publication and I will list my suggestions below.
- The authors choose to define "dementia" in a broader sense that includes Alzheimer's but is not limited to it. There is nothing wrong with this, but the title should reflect that decision. Instead of "...associated with Alzheimer's" it should be "...associated with dementia."
- The authors do address the next issue to some degree but it is a real weakness of the paper that should be addressed as a weakness in the discussion. The manuscript focuses on previously written reviews and meta-analyses, not on experimental or observational studies. As a result, the findings of one or two studies may have disproportionate influence due to the fact that they are cited in multiple reviews. To some degree the authors acknowledge this in line 291 with regard to Hayden et al; however, this is a big weakness and should be discussed at the end of the paper. The paper would be much better if the review focused on original research and observational studies rather than on other reviews which may be overly influenced by repeated citation of a small group of papers.
- The authors have put together a useful spreadsheet with which to write the paper and this is listed as Table 1. However, this spreadsheet should not be in the paper. It is not a table, just a spreadsheet tool. A table should be concise and self-explanatory. This spreadsheet must be removed. The authors could make the spreadsheet into a table if they choose to do so. If so, it must be shortened substantially by changing the content. For example, the "study" column would only need the in-text citation such as "Alizou et al, 2020". The manuscript is not publishable if the current spreadsheet is retained.
- Sometimes the results of the studies are presented in a misleading way. For instance, on line 301, the odds ratio is reported as 1.1, but without a confidence interval. This is probably not a significant increase in the odds either statistically or practically. In another example, an odds ratio of 1.5 is used to suggest that the result of exposure increases the odds of disease by 50% even though the confidence interval included "1". The level of the confidence interval was not reported. Was it 95% or 90%? This is actually pretty important in the interpretation. In general, the relative risks on most of these are pretty low and not very convincing (though there are some exceptions such as the OR reported for DDE in line 297. That OR of 4.18 with a 95% confidence interval that does not include 1 is fairly convincing). However, as with a later reference comparing organochlorine and organophosphate insecticides to other pesticides (line 320), interpretation may be important. Since almost all organochlorine and most organophosphate insecticides have been banned or replaced on the market for several years now, the people with heavier exposures to OC and OP will be older. Were these studies adjusted for age? If not, what you may be picking up in your reviews is that the exposures have changed with older people being exposed to OP and OC insecticides and the younger groups to pyrethroids. In that case your increase in dementia could be due to age rather than the chemicals in question. The cited studies need to be accompanied by a list of variables that were statistically controlled within each study. The biggest ones to report are probably age, smoking status, alcohol use, and perhaps illicit drug use, each of which is strongly associated with dementia. Otherwise, the odds ratio or relative risk is not very informative.
I will mention a few compositional issues. Do not start a sentence with an abbreviation. Do not connect words with a slash (examples "sporadic/late onset" or "substances/substance groups". These are not words and so should not be used.
I hope the authors find these comments useful.
Reviewer 3 Report
This is an well written manuscript. The study is well constructed and the findings are clearly laid out. My only main comment would be to reformat the large table (Table 2) that summarises the studies. Currently it is double spaced and appears across a number of pages. It would be much more readable if spaced similarly to the rest of the manuscript.
I wonder if the final column could perhaps be included with 'Outcomes' column information, to reduce the number of columns and allow other columns that are quite narrow to be broader. I defer to the editor's advice here.
I also wonder whether the conclusion should be in sentence form rather than dot points. I defer to the editor's advice and preference here.
Reviewer 4 Report
Thank you for the opportunity to revise this interesting review on AD and its association with environmental substances.
I have found the study design appropriate, and the discussion thorough and supported by the findings.
The manuscript falls within the aims and scopes of the Journal and is well written and presented.
Overall, I really enjoyed reading the article, and I believe it could be of interest to the readers.
I have only a minor suggestion the authors should consider. In particular, even if the study is a scoping review, there is a lack of information on the number of the literature studies selected for the review, as well as the selection criteria adopted.
With this regard, I suggest not only implementing the text of the methods section but also inserting a PRISMA-like flowchart. This latter could help the readers as it improves the clarity of the search methodology adopted.
I hope the authors will follow the suggestions as I believe the manuscript will be well received by the academic community.
Round 2
Reviewer 2 Report
I still think that Table 2 is inappropriate for a peer-reviewed paper. It is too large and is essentially the tool used to write the paper. However, I leave the decision regarding the use of Table 2 up to the editor.